# A Deep-Learning Proteomic-Scale Approach for Drug Design

**DOI:** 10.3390/ph14121277

**Published:** 2021-12-07

**Authors:** Brennan Overhoff, Zackary Falls, William Mangione, Ram Samudrala

**Affiliations:** Department of Biomedical Informatics, Jacobs School of Medicine and Biomedical Sciences, University at Buffalo, Buffalo, NY 14203, USA; brennano@buffalo.edu (B.O.); zmfalls@buffalo.edu (Z.F.); wmangion@buffalo.edu (W.M.)

**Keywords:** computational drug design, deep learning, multiscale, polypharmacology, autoencoder, docking, recurrent neural network

## Abstract

Computational approaches have accelerated novel therapeutic discovery in recent decades. The Computational Analysis of Novel Drug Opportunities (CANDO) platform for shotgun multitarget therapeutic discovery, repurposing, and design aims to improve their efficacy and safety by employing a holistic approach that computes interaction signatures between every drug/compound and a large library of non-redundant protein structures corresponding to the human proteome fold space. These signatures are compared and analyzed to determine if a given drug/compound is efficacious and safe for a given indication/disease. In this study, we used a deep learning-based autoencoder to first reduce the dimensionality of CANDO-computed drug–proteome interaction signatures. We then employed a reduced conditional variational autoencoder to generate novel drug-like compounds when given a target encoded “objective” signature. Using this approach, we designed compounds to recreate the interaction signatures for twenty approved and experimental drugs and showed that 16/20 designed compounds were predicted to be significantly (*p*-value ≤ 0.05) more behaviorally similar relative to all corresponding controls, and 20/20 were predicted to be more behaviorally similar relative to a random control. We further observed that redesigns of objectives developed via rational drug design performed significantly better than those derived from natural sources (*p*-value ≤ 0.05), suggesting that the model learned an abstraction of rational drug design. We also show that the designed compounds are structurally diverse and synthetically feasible when compared to their respective objective drugs despite consistently high predicted behavioral similarity. Finally, we generated new designs that enhanced thirteen drugs/compounds associated with non-small cell lung cancer and anti-aging properties using their predicted proteomic interaction signatures. his study represents a significant step forward in automating holistic therapeutic design with machine learning, enabling the rapid generation of novel, effective, and safe drug leads for any indication.

## 1. Introduction

Drug discovery—identifying chemicals with therapeutic effects against a particular indication/disease that is safe for human use—is a long, laborious, and expensive process. On average, $3 billion and about 15 years are required to bring a novel chemical entity to the market using traditional approaches [1]. Computational methods are a popular means of identifying potential leads through paradigms such as high-throughput virtual screening [2,3,4,5], where simulations are run to assess the binding affinity of a library of compounds against a therapeutic target of interest. The combinatorial explosion of binding poses [6,7] and ligand conformations [6,8,9] and the chaotic nature of such dynamical systems [10] prevent popular virtual screening methods from producing safe and effective therapeutic leads a priori. These issues are exacerbated by the fact that virtual screening studies usually consider a single protein target, whereas drugs ingested by humans go through absorption, dispersion, metabolism, and excretion (ADME) and exert their effects (and side effects or toxicity (T)) via interactions with multiple targets and systems [3,11,12,13,14,15]. Furthermore, the chemical space explored by virtual screening is limited to a relatively small selection of compounds when compared to the vastness of the small molecule space [4,16], thus missing more effective and safer leads.

Computational methods are efficient, accurate, holistic (i.e., take into account the entire interaction space of chemical entities), and have breadth in terms of chemical space exploration necessary to overcome the limitations of traditional approaches [2,6,12,13,17,18,19,20,21,22,23,24,25,26,27,28,29,30,31,32,33,34]. To expand compound libraries utilized in screening, combinatorial chemistry and machine-learning design pipelines have been developed to generate libraries of compounds likely to bind to a given target [35,36,37]. Some notable examples in machine learning include Insilico Medicine’s Chemistry42 platform, which designs compounds to a binding pocket [38], or a recent transformer-based network that utilized machine translation methods to generate binding ligands for the amino acid sequence of a target protein [39]. However, to take full advantage of these leads, additional screening and in vivo work must be performed to identify off-target binding as these approaches do not address the multitarget nature of drug interactions [11,13].

Various encoder–decoder models [40,41,42] for conditional [43] molecular generation on multiple properties have been proposed [17,44,45,46], but in most cases, these properties are limited to physiochemical ones. These models, however, do show great promise in their ability to rapidly generate compounds with desired properties. The most sophisticated conditional molecular generation performed thus far to our knowledge is inducing a differential expression profile of several hundred genes [17]. In all these models, proteins, the functional molecules that are primarily bound by human-ingested drugs to ensure efficacy and ADMET, remain to be considered explicitly on a large scale. Determining interactions between drug candidates and target proteins on a proteomic scale will offer the most comprehensive predictions for bioactivity and safety as many on- and off-targets will be considered simultaneously.

We developed the Computational Analysis of Novel Drug Repurposing Opportunities (CANDO) platform for shotgun multitarget drug discovery, repurposing, and design to overcome the aforementioned limitations of traditional single-target approaches [18,19,20,21,22,23,24,25,26,27,28,29]. The platform screens and ranks drugs/compounds for every disease/indication (and adverse event) through the large-scale modeling and analytics of the interactions between comprehensive libraries of drugs/compounds and protein structures. CANDO is agonistic to the interaction scoring method used; two primary pipelines within the platform allow for rapid screening and assessment of billions of drug/compound to protein interactions with fast bioanalytic docking and machine learning affinity regression protocols. Machine learning is also used to improve performance in conjunction with preclinical data in an iterative manner. Finally, CANDO implements a variety of benchmarking protocols for shotgun repurposing, i.e., to determine how every known drug is related to every other in the context of the indications/diseases for which they are approved, which enables the evaluation of various pipelines and protocols within and external to the platform for their utility in drug discovery. The multiple fast and accurate interaction scoring/docking protocols, the proteomic scale, and rigorous all-against-all benchmarking used within the platform make it unique and ideal for the design of chemical entities that target a desired proteomic space or objective.

Here, we describe the development and rigorous benchmarking of a multi-step deep learning pipeline for drug design. These pipelines perform conditional drug design given a desired proteomic interaction signature using a generative approach to explore the vastness of the entire small molecule space, while evaluating the functional behavior of candidate designs across the proteomic space. The CANDO platform’s benchmarking strategy is used in a modified fashion to determine the performance of the designed compounds relative to an objective drug. We show that the best generated designs were evaluated as being equivalent or better than a variety of controls for twenty objective drugs. Our pipeline represents a significant leap in automating holistic drug design with machine learning, with the ability to rapidly generate effective and safe drug candidates that accurately target multiple proteins within a proteome as desired.

## 2. Results and Discussion

### 2.1. Behavioral Similarity of Designed Compounds to Their Objectives

We observed excellent performance of our Reduced Conditional Variational Autoencoder (RCVAE) proteome-scale design pipeline in all our benchmarking experiments following training (see the methods Section 3.2). If this performance continues to hold following synthesis and validation, it indicates that this pipeline will greatly enhance the pharmaceutical discovery pipeline for novel treatments against a variety of simple and complex indications. A critical aspect of verifying the utility of the designs generated was to compare their predicted behavior (i.e., proteomic interaction signatures) to their intended behavior, which was input to conditional generation. If the predicted behaviors of designed compounds were highly similar to the conditional objective across objectives relative to the corresponding controls, we concluded that the RCVAE design pipeline may be used to accurately design compounds that possess any desirable bioactivity and subsequent function, given the extensive benchmarking and validation the CANDO paradigm has undergone [18,21,26,29,47,48,49,50]. This is the primary motivation and goal for using the CVAE architecture in terms of accelerating drug discovery: design with respect to arbitrary numbers of on-, off-, and anti-targets (Figure 1).

We evaluated the performance primarily by the median of each distribution, indicated by the horizontal bars in the box plots in Figure 2. Lower root-mean-squared deviation (RMSD) values indicate a greater reproduction of proteomic interaction signatures for the intended and/or predicted behavior of any given compound, i.e., greater behavioral similarity. Every set of redesigns performed significantly better (*p*-value of ≤ 0.05) than a selection of random compounds according to this criterion. Additionally, despite being comparatively close in predicted proteomic behavior, our redesigns maintained high levels of structural diversity, as evidenced by their Tanimoto coefficients to our drug library (average ≤ 0.39). For sixteen of the objectives (excluding sirolimus, cucurbitacin Q1, digoxin, and myriocin), the redesigns significantly outperformed the corresponding top 100 and same indication controls. The top 100 control, which included several “me too” compounds (or structural analogs) for each objective [24,51], is the most rigorous one we could devise and illustrates that just generating 100 designs in many instances produces more behaviorally similar compounds to a desired interaction signature than selecting the most similar 100 compounds from a total of 13,194 (the size of the CANDO drug library). The existence of structural analogs in the top100 control indicates bias in favor of already effective compounds in an effort to break into a new market or retain market dominance by generating new intellectual property. New drugs are often derivatives of existing ones with small changes, which our design pipeline is able to overcome, particularly with a bit of extra effort (see Section 2.4 below). Overall, these results indicate that the RCVAE design pipeline produces compounds that accurately match the behavior of desired proteomic interactions relevant to drug discovery. In other words, interactions related to therapeutic efficacy, ADME, and toxicity are modulated precisely in the designed compounds, particularly for objectives from the rational sources subset.

### 2.2. Relative Performance Gains of Designs Relative to Controls

The bottom panel of Figure 2 compares the relative performance gains for proteomic objectives from the rational design and natural sources subsets relative to the controls. Compounds in the natural sources subset were derived from massively parallel evolutionary processes over eons of time. As a result, they exhibit evolutionary drift, resulting in complex behaviors that are suboptimal from a therapeutic discovery perspective, i.e., unnecessary off-target interactions and/or individual interactions drifting away from functional free energy minima [52]. In addition to verifying that our designs behave as intended, our benchmarking also shows that the RCVAE design pipeline accomplishes this replication of behavioral similarity through an abstraction of rational drug design. That is, the similarities of redesigns to objectives from the rational design subset were greater relative to those from the natural sources subset (Figure 2). Adopting an abstraction of rational drug design is optimal for the impact of this platform on drug discovery because if the model was merely replicating the molecular structure of design objectives and not proteomic behavior specifically, the limitations of natural products (low synthetic accessibility, poor ADMET [53]) would present themselves in the designs in addition to indicating that the model may be over-trained. This discrepancy in the similarity of redesigned natural products and rationally designed drugs, therefore, further supports the notion that the RCVAE pipeline is able to intelligently design compounds with desirable bioactivities across multiple targets.

**Figure 2 pharmaceuticals-14-01277-f002:**
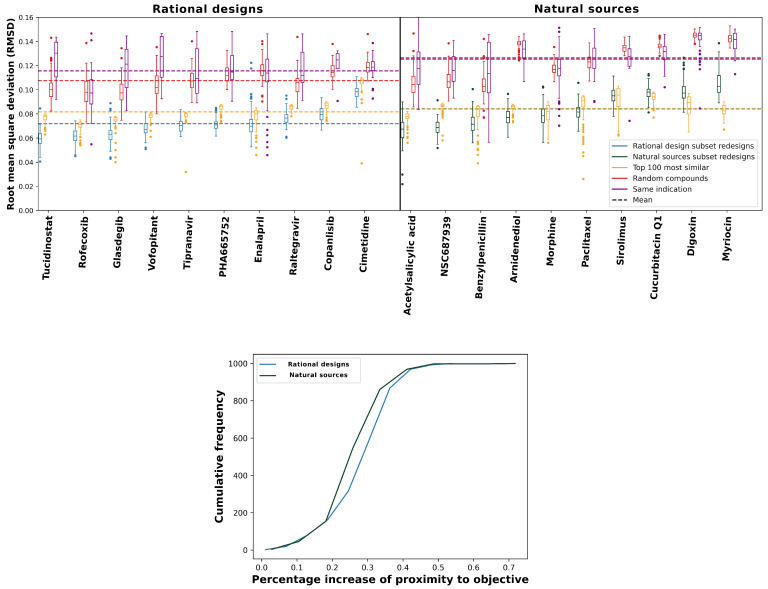
**Performance of our deep learning drug design pipeline.** To evaluate the performance of the RCVAE pipeline for drug design, we compared redesigned compounds to three controls. The root-mean-squared deviations (RMSDs) between redesigned and control interaction signatures for the rational design and natural sources subsets and the corresponding objective signatures were used to evaluate the performance, as a proxy for behavioral similarity. In the top panels, the blue and green box plots denote the distributions of RMSDs between predicted proteomic interaction signatures for 100 redesigns and the ten corresponding objectives from the rational design (left) and the natural sources (right) subsets along the horizontal axis, respectively. For each RMSD distribution box plot, the boxes indicate the first and third quartile ranges, the horizontal bar indicates the median, and the whiskers indicate non-outlier ranges, with outliers plotted as dots. As a naive control, the red box plots in both panels denote the distributions of the RMSDs between predicted proteomic interaction signatures for a set of 100 randomly selected drug-like compounds taken from the ZINC database [54] and the corresponding objectives. Yellow box plots denote the distributions of the CANDO-predicted top 100 most similar compounds by interaction signature and their corresponding RMSDs; this is a more rigorous control, which checks to see if there exists any compound in the CANDO library that could match or exceed the performance of the design pipeline. Purple box plots denote the RMSD distributions of compounds approved for the same indication as the objective; this is a third control based on phenotype. Dots represent RMSD values for outlier points for each distribution. Dashed lines show the average RMSDs of all compounds corresponding to a given color. All redesign–control distribution pairs were significantly different as indicated by Kolmogorov–Smirnov (K-S) tests and outperformed random controls. For 16/20 of the objective compounds, our redesigns were able to perform better than the top 100 and same indication controls (medians and distributions). The exceptions were four compounds from the natural sources subsets (sirolimus, cucurbitacin Q1, digoxin, and myriocin), discussed further in Section 2.2 and Section 2.4. The bottom panel displays cumulative frequency graphs of percentage increases of similarities (“proximity”) to the objective for rational design and natural sources subsets redesigns (see the Materials and Methods Section 3.3). Redesigns from the former subset were significantly more accurate to the objective compound than the natural sources subset when compared to the naive control, with mean percentage similarity increases of 33.4% and 32.9%, respectively (*p*-value ≤$7.0\times 10^{-6}$). This indicates that not only does the design pipeline generate compounds with the desired proteomic-scale behavior, but that it likely implements a learned abstraction of rational drug design.

### 2.3. Visualizing and Filtering Using t-SNE Plots

For all objectives, the redesigns greatly outperformed existing drugs approved for the desired indication when compared to the objective signature. Despite this, some Kolmogorov–Smirnov (K-S) tests indicated weaker statistical significance when differentiating between the RCVAE design and the same indication distributions. To understand the distributions of redesigned and control compounds and to provide an additional filtering mechanism to evaluate the performance of our designs as illustrated in Figure 2, we visualized the interaction signatures of all compounds (objectives, redesigns, and all controls) evaluated in our benchmarking with t-SNE plots [55] (Figure 3).

t-SNE plots generally corroborate the relative behavioral similarities between redesigns and controls relative to their objectives. RCVAE pipeline designs tend to cluster densely around the objective compound, as they are designed to do. The top 100 compounds cluster around these designs, with a few structural analogs very close to their objectives. Finally, the same indication and random compounds clustered the farthest from the objective. As the same indication compounds represent a group of diverse drugs approved for an indication that includes the objective, there is a general lack of clustering, and they are at greater distances from their objectives. Comparing the performance of the designs relative to their objectives and controls using both Figure 2 and Figure 3 indicates that the t-SNE plots illustrated in the latter may be used to assess the confidence in and room for improvement of the design pipeline performance for specific objectives.

### 2.4. Improving Cases with Sub-Optimal Performance

For eleven objectives, a handful of outliers in the top 100 or same indication controls outperformed the design distributions (Figure 2 and Figure 3). As noted above, structural analogs create a bias when evaluating performance due to them having very similar behavioral interaction signatures. We observed that the top 10 compounds (out of the top 100 controls, covering almost all outliers) yielded an average Tanimoto coefficient of 0.51, in contrast to an average of 0.39 for all designs, relative to their objectives. We further observed that 23/200 top 10 compounds, across all 20 objectives, had a Tanimoto coefficient ≥0.90 with an average of 0.96, demonstrating the “me too” bias with the top 100 control. Unlike the few top 100 outliers, the designs offer a larger and more diverse selection pool for prospective validation.

Regardless, we further investigated the behavior of the RCVAE pipeline for one of the objectives (cucurbitacin Q1) where an outlier was clearly better than the best designed compound by expanding the number of designs generated by the RCVAE pipeline from 100 to 1000 compounds. We found that the RMSDs of the top 100 out of 1000 designs, far less than the 13,194 compounds that were the source for the top 100 most similar compounds control, ranged from 0.073 to 0.09. This placed the RMSD of the best designs well below the lowest RMSD outlier of the top 100 control. Altogether, this indicates that the performance of the RCVAE pipeline may be enhanced by increasing the number of designs generated and selecting for behavioral similarity. Our design pipeline offers structurally diverse lead compounds with the potential to match or exceed the behavioral similarity of the top CANDO predictions from its known drug library for noisy objectives, such as compounds from the natural sources subset.

### 2.5. Synthetic Feasibility of Designed Compounds

To viably demonstrate that our 2000 redesigns were synthetically feasible, we utilized a high-throughput machine-learning-based approach to predict synthetic complexity scores, called SCScore. SCScore utilizes a database of known synthetic reaction pathways for training to make predictions of how easy or difficult it is to synthesize a novel compound.

As shown in Figure 4, predicted synthetic complexities for the RCVAE pipeline redesigns are often comparable to their design objective, i.e., the objective compound’s scores exist within the distribution of the scores for the redesigns. Low Tanimoto coefficients (average 0.39) between designs and objective compounds indicate that the comparable synthetic complexities of our redesigns are not due to a corresponding high structural similarity. In other words, despite the high structural diversity of the generated designs, the synthetic complexity for objective and designed compounds remains stable. This may be explained by the maintenance of functionally relevant substructures of comparable synthetic complexity that are present in any given objective and redesign that are combined differently to produce similar behaviors and low structural similarity. We are studying this phenomenon more thoroughly with larger datasets by investigating the redundancy of the substructures of all approved drugs and redesigns for publication in a future study.

Additionally, predicted synthetic complexities are typically lower for objectives and redesigns in the natural sources subset relative to the rational design one, indicating that the SCScore software may be biased by what is already well known. As objectives in the natural sources subset score similarly to their redesigns, it is somewhat plausible that evolutionary optimization towards synthetic accessibility, at the expense of macromolecular interaction free energies [52], may account for the diminished behavioral similarity of RCVAE designs between objectives in the rational design and natural sources subsets correspondingly (see Figure 2). A comprehensive analysis of this hypothesis with larger subsets is necessary to validate or falsify this hypothesis.

It is useful to note that many RCVAE designs are more synthetically accessible than the compound whose behaviors they replicate. As Figure 2 already indicates high behavioral similarity between redesigns and their objectives, the RCVAE design pipeline may serve the additional purpose of designing analogs to existing drugs that are more synthetically feasible and therefore easier and less costly to produce. This is accomplished without adding a synthetic complexity parameter to the condition vector for compound generation (Figure 4).

Finally, we routinely used several other methods to evaluate RCVAE designs such as the Quantitative Estimation of Drug-likeness (QED) [57], Synthetic Accessibility Score (SAScore) [58], and AiZynthFinder [59] to assess their chemical viability and drug-likeness. We also compared our designs to benchmarks from GuacaMol [60]. These results corroborated the outputs of our benchmarking and/or SCScore; future work will include a rigorous evaluation of drug design technologies, much as we have done for repurposing [29].

### 2.6. Applications to Aging and Non-Small Cell Lung Cancer

A fundamental tenet of CANDO is that evaluating all the possible interactions between a human-ingested drug/compound and the macromolecules and systems it encounters on a proteomic/interactomic scale is necessary to determine its safety and efficacy for a given indication [18,19,20,21,22,23,24,25,26,27,28,29]. The RCVAE design pipeline represents a significant step forward in early drug discovery as it allows for the virtually unlimited generation of novel putative drug candidates to treat any indication/disease by combating it on a proteomic scale. As a precursor to upcoming work, we generated designs for several objective compounds/drugs associated with human or cell longevity (“aging”) [61,62,63,64,65,66,67,68,69,70] and approved for Non-Small Cell Lung Cancer (NSCLC) [71,72,73,74,75,76,77,78]. We expect our pipeline to produce redesigns that retain the proteome-scale behaviors of these objectives used to treat these complex indications/diseases, while being structurally diverse.

Figure 5 illustrates the top redesigns for the thirteen objectives covering the two indications ranked using two metrics based on the greatest similarity criterion: lowest RMSD between corresponding interaction signatures and highest Tanimoto coefficient between corresponding molecular fingerprints. Two classes of redesigns, all with the greatest behavioral similarity to their objectives, emerged when performing the comparisons illustrated in Figure 5: designs that chemically/structurally resemble their objective compound/indication (metformin, NAD+, resveratrol, curcumin, and RepSox for aging and gefitinib, erlotinib, afatinib, and dacomitinib for NSCLC, all with a Tanimoto coefficient ≥0.39) and those that do not. The former class is intriguing as it implies the existence of highly optimized structures for a given phenotype (indication), as shown by the convergence of redesigns to known drugs. It also demonstrates that the RCVAE design pipeline produces highly similar designs to known drugs to perform specific tasks, only from their proteomic interaction behavior and without being exposed to any similar structures in training. In other words, the proteomic-scale interaction information for a compound is enough information for our design pipeline to reliably reconstruct a chemically/structurally similar compound in some cases. The latter class demonstrates the expanse and diversity of a chemical space not yet charted by medicinal chemists, capable of replicating the interactions of known drugs. We are in the process of synthesizing the top designs from these pipelines for these indications and validating them in corresponding preclinical models via industry partners and collaborators.

### 2.7. Limitations and Future Work

To treat an indication in a comprehensive fashion, especially complex ones such as aging or NSCLC, entirely novel drugs will likely need to be developed that go beyond replicating the systemic effects of existing ones. This would require a thorough and accurate description of the interaction networks responsible for disease etiology, as well as compound behavior to ensure optimal efficacy and safety. The creation of these interaction networks may be accomplished by multiscale modeling, literature/database analyses, and/or high-throughput experimental studies, all of which may be incorporated within CANDO. We are currently in the process of conditioning the RCVAE design pipelines and comparing them to ones based on graph neural networks [79] using these more complex interaction networks that go beyond the information present in the linear signatures.

**Figure 5 pharmaceuticals-14-01277-f005:**
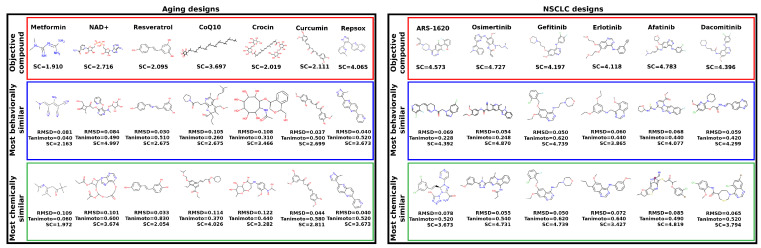
**Analysis of the top designs generated for aging and Non-Small Cell Lung Cancer (NSCLC).** The names, chemical structures, and predicted Synthetic Complexity (SC) scores of the objective compounds for aging (left) and NSCLC (right) are displayed in the top row of each panel (red border). The chemical structures of the most similar redesigns to each objective based on their interaction signatures alongside three metrics for these redesigned compounds relative to their objectives (RMSD of interaction signatures, Tanimoto coefficient, and predicted synthetic complexity) are displayed in the row below (blue border). The chemical structures of the most similar redesigns using molecular fingerprints alongside the same three metrics (RMSD, Tanimoto, and synthetic complexity) for these redesigns relative to their objectives are displayed in the bottom row (green border). The designs generated using the RCVAE pipeline retain a fair amount of structural diversity, as indicated by the fingerprint comparison scores (low Tanimoto coefficients) whilst maintaining high predicted behavioral similarity in terms of interaction signatures (low RMSDs). Other methods for evaluating our designs such as QED corroborated the above results [57]. These designs demonstrate the utility of our pipeline for designing novel compounds to combat complex indications on a proteomic scale and may be pursued further via preclinical validation studies.

We are currently exploring other scoring protocols within the CANDO platform for conditional generation to overcome the limitations of any specific interaction calculation method. For example, the interaction scoring protocol used in this work has been shown to have great utility in the context of evaluating proteomic behavioral similarity based on benchmarking performance [18,19,20,21,22,23,24,25,26,27,28,29]. However, information on agonism/antagonism, and downstream functional activity upon binding may be obtained via design pipelines that utilize gene and protein expression data, which we are incorporating into CANDO. Public gene expression data, available through the L1000 and Connectivity Map projects [80,81], highlight important genes/proteins that are upregulated and downregulated following exposure to a drug/compound. For example, if gene expression data suggest that certain downstream genes are significantly upregulated in a given pathway following interaction with a compound and that same compound is predicted to have strong interactions with multiple proteins in that pathway, it can be inferred that those are likely to cause activating/agonist behavior. The same can be inferred for downregulated genes and inhibition/antagonism. In addition, we are exploring the use of high-throughput robotic systems such as DESI-MS to generate large-scale interaction and activity data [82,83,84,85,86,87,88,89,90]. CANDO thus enables and illustrates the benefit of combining heterogeneous sources (gene, as well as protein expression, protein pathway databases, high-throughput binding, and activity data) to create novel types of interaction signatures/networks to produce design objectives that tackle complex indications.

The CANDO platform enables the benchmarking of any arbitrary proteome/protein library for its utility in drug discovery using a similar all-against-all process as described in Section 3.3. The generation of highly accurate modeled protein structures such as those predicted by Deepmind’s AlphaFold offer an attractive representation of the full human proteome to perform such benchmarking, which we have completed and will publish separately. This allows for conditional drug design using AlphaFold interaction signatures, especially giving us greater coverage and control over particular proteins and pathways to modulate with expert input for specific indications.

The benchmarking and performance evaluation of our RCVAE drug design pipeline were based on using known data as the ground truth or gold standard. While computational experiments are an important first step and indicate promising, prospective preclinical validation of the pipeline, its designs will require medicinal chemistry synthesis, binding studies, and disease models assays at multiple scales, which we are currently undertaking. Regardless, our combined work to date [18,19,20,21,22,23,24,25,26,27,28,29], including this study, indicates that novel high-throughput methods for rapidly identifying relationships between compounds, proteins, pathways, and cells are highly desirable for holistic drug discovery. As CANDO is agnostic to the specific methods used for any of its protocols, should such data become available, the RCVAE design pipeline described here would be well poised to take advantage of them for maximum drug discovery efficiency.

## 3. Materials and Methods

Figure 1 illustrates our overall methodology to create a new drug design pipeline. We employed the Computational Analysis of Novel Drug Opportunities (CANDO) platform to generate proteomic interaction signatures for the compounds in the training set of our learning-based model. The CANDO interaction signatures had their dimensionality reduced in an autoencoder, which models the underlying correspondence between protein structures as they function in the proteome. The reduced signatures were then used as labels for each training molecule, which the generative Conditional Variational Autoencoder (CVAE) model learns to reconstruct given a target interaction signature. We then used CANDO to benchmark the performance of the designed compounds in the context of their objectives and make predictions of novel designs for two indications for future prospective validation.

### 3.1. Compound–Proteome Interaction Signature Generation Using the CANDO Platform

Multiple pipelines for multiscale therapeutic discovery, repurposing, and design have been implemented in the CANDO platform [18,19,20,21,22,23,24,25,26,27,28,29]. Here, we utilized CANDO to simulate the interactions between a given drug/compound and a library of protein structures to generate the corresponding proteomic interaction signature.

The protein structure library used in this study is a set of 14,606 nonredundant structures derived from the Protein Data Bank (PDB) [91] corresponding to the human proteome fold space (“nrPDB”) [92,93,94,95,96,97]. The compound–protein interaction scores in these signatures are computed using the bioanalytic docking protocol BANDOCK, which compares query compound structures to all ligands that are known or predicted to interact with a protein binding site [22,27,29].

Potential binding sites on a protein are elucidated using the COACH algorithm, which uses three different complementary algorithms and a consensus approach to consider the sequence or substructure similarity to known PDB binding sites [98]. COACH has been utilized extensively within the CANDO platform to accurately predict the binding behavior of numerous compounds against numerous targets, as demonstrated by its benchmarking performance in multiple studies (Section 3.3 and [18,19,20,21,22,23,24,25,26,27,28,29]). For each potential binding site, the COACH output includes a set of co-crystallized ligands, which are compared to a compound of interest using binary chemical fingerprinting methods that describe the presence or absence of particular molecular substructures [99]. The maximum Tanimoto coefficient between the binary fingerprints of the query compound and the set of all predicted protein binding site ligands becomes the interaction score. The better the score, the higher the likelihood of the interaction being correct due to the inferred homology. Thus, if there are proteins with multiple binding sites and corresponding ligands, the strongest interaction is used. If there are no matches, then the score returned is zero (i.e., no interaction). The final output is a vector of 14,606 scores comprising the interaction signature between a given compound and the nrPDB library. Further detail on the pipelines used to generate and benchmark the interaction signatures is given elsewhere in numerous publications [18,19,20,21,22,23,24,25,26,27,28,29].

### 3.2. Model Architecture and Data Generation

We selected a CVAE [45] architecture for generating novel molecular structures. Training data consisted of SMILES [100] strings labeled with predicted proteomic interaction signatures based on the nrPDB library. The training set consisted of 300,000 compounds selected at random from the ZINC database [54]. The 14,606 protein binding scores were predicted for each compound. The dimensionality of the protein interaction signatures was reduced to a 200-dimensional vector via a conventional autoencoder [101]. Following an input layer with 14,606 neurons, the encoder consisted of 10 sequential, densely connected layers with 10,000, 7750, 5500, 2250, 2000, 1250, 1000, 500, 250, and 200 neurons in each layer, respectively. This was reversed in the decoder, and a final layer with 14,606 neurons was used as the output to the network. The root-mean-squared deviation (RMSD) between the input and reconstructed signatures was used as a loss metric. This model was trained on 250,000 compounds until over-fitting was observed, which occurred after 15 epochs/iterations. Each epoch was validated on another 50,000 randomly selected compounds. The model was then used to reduce the non-redundant signatures of the training compounds. These became the labels for each SMILES string present in the CVAE training data.

Before being input into the CVAE, SMILES strings were one-hot encoded [102,103], resulting in a rank-2 tensor of size (sequence length × vocab length), where sequence length is the maximum number of characters allowed per SMILES string and vocab length is the unique number of characters represented in the input data. The reduced protein signature, *c*, was appended to the end of the one-hot encoding repeated at each sequence position (commonly referred to as time steps in the context of Long Short-Term Memory (LSTM) cells [102,104,105]). Similar to [45], the tensor was fed sequentially through three LSTM cells [104,105] to encode the original input. The encoder outputs to two parallel layers, one representing the mean and one for the standard deviation of the latent vector. The latent vector, *z*, consists of 200 dimensions and is sampled from the encoder output. The latent vector is then repeated for the total number of time steps, and the protein signature, *c*, is re-appended onto the resulting tensor in the prior fashion. This is input into the decoder, which also consists of three LSTM cells. Finally, this is output to a matrix of probabilities for each character in a SMILES string. Taking the maximum probability token for each character slot, one-hot encoded SMILES strings denoting reconstructions of input compounds are generated.

The loss metric used to train the CVAE is as follows:(1)$$E\left[log\left(P\left(X|z,c\right)\right)\right]-D_{KL}\left[Q\left(z|X,c\right)||P\left(X|z,c\right)\right]$$
where *E* denotes the reconstruction error and $D_{KL}$ denotes the relative entropy or the Kullback–Leibler divergence [106]. P(X|z,c) denotes the probability density function approximated by the decoder for each character in a SMILES string given the latent and conditional vectors. Q(z|X,c) denotes the probability density function approximated by the encoder given the input SMILES strings and condition vector. The CVAE was trained on 300,000 compounds until convergence. The Reduced CVAE (RCVAE) model pipeline is depicted visually in Figure 1.

### 3.3. Benchmarking and Analysis of the RCVAE Design Pipeline Performance

The fundamental supposition and result of benchmarking the CANDO platform is that compounds with similar interaction signatures will behave similarly. On a proteomic scale, these behaviors take the form of efficacy and ADMET for a given indication. The CANDO platform was benchmarked using known drug-indication associations [18,19,20,21,22,23,24,25,26,27,28,29] derived from the Comparative Toxicogenomics Database [107] and, more recently, drug-adverse events obtained from OFFSIDES [108,109] and SIDER [109]. We recently published the best metrics to use for benchmarking drug repurposing platforms [29]. The results of benchmarking and prospectively validating CANDO indicate that the proteomic-scale interaction modeling of drugs elucidates their behaviors, and these behaviors correspond to treatments for indications for which these drugs are approved [18,21,26,29,47,48,49,50]. The benchmarking of the platform using known associations in this comprehensive all-against-all manner enables us to assess the correctness and utility of other parameters, such as the protein library composition, solved vs. modeled structures, different molecular docking and machine-learning algorithms, etc.

We performed several benchmarks of the RCVAE to verify its utility and robustness beyond that of its performance based on metrics used in training. Our benchmark set consisted of twenty approved or experimental objective drugs, comprised of subsets of ten derived from rational design and natural sources, respectively (Figure 2). These compounds and all related ones with a Tanimoto [110] coefficient ≥0.9 were omitted from training. Proteomic interaction signatures were computed by CANDO, and output SMILES strings were generated by the RCVAE design pipeline as described above. This resulted in SMILES strings corresponding to 100 novel redesigns for each objective drug. One-hundred compounds were selected at random from the curated ZINC database to serve as a naive control. As a second, more rigorous control, the CANDO platform was used to generate the 100 most similar compounds (“top 100”) from its 13,194-sized library to each objective drug according to their proteomic interaction signatures. As a third control, the CANDO platform’s indication prediction pipeline was used to predict a set of compounds for the indication associated with each objective, i.e., the indication that the objective is approved for (“same indication”) [27,107]. Proteomic interaction signatures were generated for all compounds (designs and controls), which were then compared to that of the objective using the RMSDs between them. The RMSD distributions are illustrated using box plots with boxes depicting the first and third quartile ranges, horizontal bars depicting the median, whiskers representing non-outlier ranges, and outliers explicitly plotted (Figure 2, Results Section 2.1).

Kolmogorov–Smirnov (K-S) tests [111,112] were used to demonstrate statistical significance [113] between samples for each redesign–control RMSD distribution pair for each objective. We also compared the performance of the RCVAE design pipeline between objectives from the rational design and natural sources subsets, respectively. To do this, we computed the average RMSD between each naive control and the corresponding objective and compared this to the RMSD of each redesign for an objective. This yielded a percent increase of similarities to the objective for each redesign given by:(2)$$P=\frac{<RMSD_{control}>-RMSD_{redesign}}{<RMSD_{control}>}$$
where *P* denotes the percent increase (“proximity”), $<RMSD_{control}>$ denotes the average RMSD between the naive control and corresponding objective, and $RMSD_{redesign}$ denotes the RMSD of the redesign when compared to the objective proteomic interaction signature. This yielded 1000 total values of percent proximity increases for both the rational design and natural sources subsets. The values for both subsets were then averaged and compared using K-S tests (Figure 2, Results Section 2.1).

To better visualize the distributions of our redesigns and controls, t-distributed Stochastic Network Embedding (t-SNE) plots [55] that show the clustering of similar interaction signatures in two dimensions were generated with the interaction signatures of all objective, redesign, top 100, and same indication compounds (Figure 3, results Section 2.3).

We also computed Tanimoto coefficients for all redesigns relative to their respective objective compounds to determine structural diversity. Finally, we utilized SCScore [56], a machine-learning platform to predict the synthetic complexity of our redesigns in relation to the corresponding objectives (Figure 4, Results Section 2.5).

### 3.4. Generating Novel Designs for Prospective Validation

Design objectives for benchmarking were selected from a diverse set of indications and approved/experimental statuses to ensure broad coverage of the proteomic interaction signature space and to mitigate potential bias in the results. Regardless, the benchmark set was repeatedly used to parameterize the pipeline described here, which has the potential to lead to overtraining. To address this and also to apply our design pipelines to relevant real-world problems of sufficient complexity where the proteomic approach would be relevant, we selected 13 (7 + 6) objective compounds that have shown promise for aging/developmental intervention [114,115,116,117,118] and NSCLC [118,119,120] to redesign for prospective validation (Figure 5, Results Section 2.6).

## 4. Conclusions

We utilized the RCVAE pipeline within the CANDO platform to take advantage of multiscale compound–proteome interaction modeling and develop an attractive approach to holistic drug design. We compared the predicted behaviors of the designed compounds to those of known drugs/compounds and demonstrated that the RCVAE pipeline is capable of generating novel compounds with the desired specificity of binding on a proteomic scale. We additionally demonstrated that compounds designed by our pipeline maintained reasonable predicted synthetic complexities and were structurally diverse. We expect the compounds designed using our pipeline for aging/developmental intervention and NSCLC will serve as novel leads for safe and effective therapeutics following prospective validation. The RCVAE design pipeline generates novel compounds that are synthetically feasible and behaviorally desirable, simultaneously taking efficacy and ADMET into account by examining interactions on a proteomic scale, which is necessary to understand the science of small molecule behavior and apply it to holistic therapeutic discovery.

## Figures and Tables

**Figure 1 pharmaceuticals-14-01277-f001:**
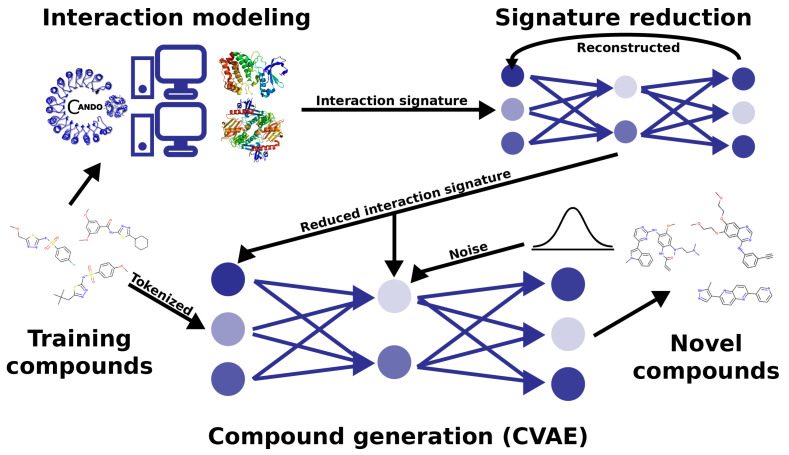
**Deep -learning architecture and pipeline for generative drug design**. We used the CANDO platform to predict interaction signatures for each compound in a training set against a library of nonredundant protein structures representing the human proteome. The interaction signatures have their dimensionality reduced in an autoencoder, which models the underlying correspondence between protein structures as they behave in the proteome. The reduced signatures are then used as labels for each training compound, which the generative conditional variational autoencoder model learns to reconstruct given a target interaction signature. This pipeline allows us to redesign behaviorally similar compounds to existing drugs based on their interaction signatures, as well as to modulate interactions on a proteomic scale as desired to generate behaviorally novel therapeutics.

**Figure 3 pharmaceuticals-14-01277-f003:**
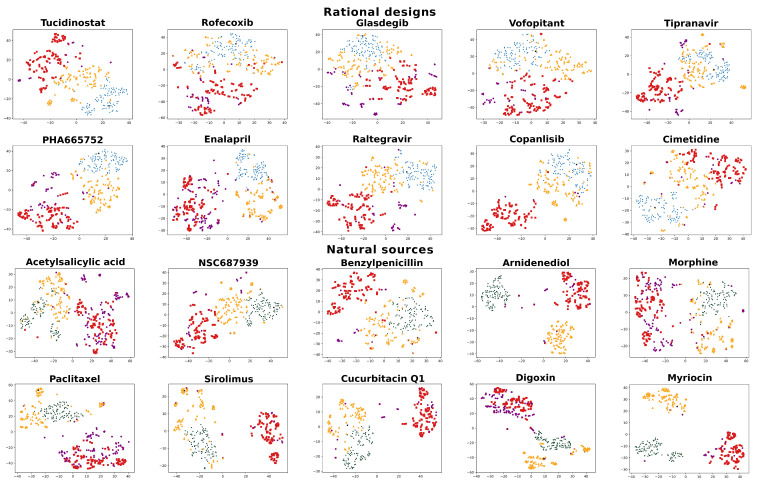
**t-SNE visualizations for the interaction signatures of objective compounds, redesigns, and various controls with structural comparisons.** t-SNE plots were generated for each of the twenty design objectives. The t-SNE algorithm was run on the interaction signatures of the objectives (black stars) and 100 redesigns (blue and green circles), as well as the 100 random (red squares), top 100 (orange triangles), and same indication (purple hexagons) control compounds. (Color coding is the same as in Figure 2.) Euclidean distances shown in t-SNE visualizations generally corroborate our findings from Figure 2 when considering the RMSD due to the maintenance of proximal points and distributions. The exceptions to this were for benzylpenicillin and paclitaxel, as Figure 2 shows greater proximity for designs than the top 100 control, whereas t-SNE plots show better clustering for the top 100 set around the objective. The generally (14/20) greater proximity of designed compound clusters to the objective point when compared to top 100, same indication, and random control compounds corroborate the behavioral similarity of designed compounds to their objective in the predicted interaction space, i.e., designed compounds are predicted to behave in the manner in which they were designed.

**Figure 4 pharmaceuticals-14-01277-f004:**
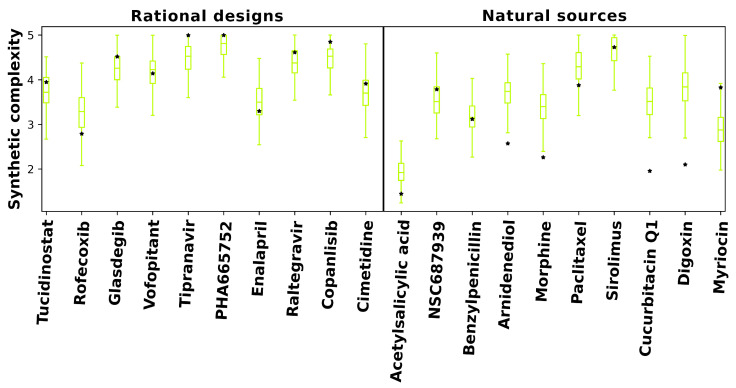
**Predicted synthetic complexity of designs compared to objective compounds. **Synthetic complexity scores for objective and RCVAE pipeline designed compounds as predicted by SCScore [56] serve as the basis for this comparison. Box plots denote the distributions of predicted synthetic complexity scores for RCVAE redesigns of objective compounds (stars). The scores of the objectives enable us to evaluate the synthetic feasibility of the redesigns relative to their corresponding objective compounds. There is a greater occurrence of low-scoring natural sources objectives in predicted synthetic complexity. This indicates that the SCScore software is biased by what is already well known or it may also highlight a potential reason why RCVAE performs better for objectives from the rational design subset: it is somewhat plausible that evolution optimizes for synthetic accessibility over binding free energies, hence the discrepancies in behavioral similarities between the designs and their objectives for the two benchmarking (rational design and natural sources) subsets (Figure 2). RCVAE designs are of similar synthetic complexity to their corresponding objective compounds as most objective complexities lie within the box plot whiskers (not outliers), with some designs being more accessible than their objectives.

## Data Availability

All relevant data are presented/illustrated in this manuscript.

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
