# Peer review of "A Deep-Learning Proteomic-Scale Approach for Drug Design"

_pharmaceuticals, 2021, doi:10.3390/ph14121277_

Round 1

Reviewer 1 Report

Author have performed deep learning based autoencoder to first reduce the dimensionality of CANDO computed drug-proteome interaction signatures. We then employed a reduced conditional variational autoencoder to generate novel drug-like compounds when given a target encoded “objective” signature. Using this model, we designed compounds to recreate the interaction signatures for twenty approved and experimental drugs and showed that 16/20 designed compounds are predicted to be significantly (p-value ≤ .05) more behaviorally similar relative to all corresponding controls, and 20/20 are predicted to be more behaviorally similar relative to a random control. We further observed that redesigns of objectives developed via rational drug design perform significantly better than those derived from natural sources (p-value ≤.05), suggesting that the model has learned an abstraction of rational drug design. 1. The current abstract is poor. No focus point in the abstract section. 2. Many short forms are used without proper explanations. 3. Currently, there are no approved drugs, and several attempts have been made to use computational program approaches in drug repurposing for COVID-19 treatment. Is this okay? 4. The purpose and significance of this research must be explained in the abstract more clearly. 5. Change the entire abstract section and make it more attractive. 6. Increase the sentences in the conclusion section. 7. Why there are no methods for in silico work? 8. The first paragraph of the introduction section contains no new information. Need to change. The paragraphs are not logically arranged; there are unnecessary repeats. Some of the explanations should be marked on their first appearance. 9. The introduction section seems missing important information. This section needs profound modification with different medical hypotheses recommended different treatment approaches and gave convincing arguments for new discoveries urgently needed for treatment option COVID-19. Authors can also write some about the recent mutation. 10. In silico prediction of a drug is unlikely fitting with the wet-lab data. Experimental validation is highly required to recommend any predicted compounds as a potential vaccine/drug. 11. Author did not discuss the structural relationship of the compounds in the observed results. Such as which skeleton/functional groups/chemical class most probably contributes to observed binding affinities, etc., by citing appropriate references. 12. Author should provide critical justifications of the observed results. 13. It is highly recommended to emphasize findings and assumptions that support or disagree with other work(s). 14. Addendum, repetition of the results should be avoided in the discussion section. 15. A sound discussion includes principal, relationships, and generalizations supported by the results. 16. Additionally, I am confused about the discussion. It does not seem to really discuss the data that was described in the manuscript. I would suggest that the authors refocus their discussion to clarify how the results of their work fit into the larger picture of what is current today instead of describing more of a literature background. 17. Author should stress the significance of the study. Novelty of the work should be supplemented by the author (in the conclusion section). 18. This section should be supported by the results/insights. Conclusively, it will confer a distinct idea of the study. General Comments: 1. English is weak. The authors need to improve their writing style. The whole manuscript needs to be checked by native English speakers. 2. Every section of the manuscript must be written scientifically according to the published literature with appropriate references. 3. The logical flow of this manuscript is not perfect. The authors have written several matters haphazardly. 4. The work appears as groundwork. 5. Spacing, punctuation marks, grammar, and spelling errors should be reviewed wholly. 6. The study problem is not obvious.

Reviewer 2 Report

The paper by Overhoff et al with the title "A Deep Learning Proteomic Scale Approach for Drug Design" describes a machine-learning-based approach based on the analysis of drug-target interaction signatures based on the computational analysis of novel drug opportunities (CANDO) platform.

The purpose of the paper is highly ambitious.

The pipeline is well explained and so is the overall methodology. A validation stage is introduced. However, the model has in my view several limitations emerging from the models/approximations used. While these represent a natural cost of such a large scale approach, they are not mentioned at all. These potential limitations should be well discussed and justified.

Examples:

Line 120. "The protein structure library  used in this study is a set of 14606 nonredundant structures derived from the Protein Data Bank (PDB) corresponding to the human proteome fold space (“nrPDB”) "

The human proteome is much larger than what is represented in the PDB. The AlphaFoldDB has now high quality predicted structures for the entire human proteome. The authors should justify this imbalance between the number of proteins considered and the human proteome and the possible implications/limitations of the results.

Line 122. "The compound-protein interaction scores in these signatures are computed using the bioanalytic docking protocol BANDOCK that compares query compound structures to all ligands that  are known or predicted to interact with a protein binding site "

For many human proteins in the PDB no ligands are currently co-crystallized. How is this taken into account? And binding affinity?

Line 123. The coach algorithm should be better described here with indication of its accuracy in representative test sets.

Has any structured-based docking confirmation to any of these targets be attempted to validate the results? With specific protein-ligand docking tools such as GOLD or AutoDOCK?

In the human proteome some binding sites are associated with antagonists, other with agonists. How is this taken into account?

How about proteins with multiple possible binding sites, depending with the characteristics of the potential molecule? How is this accounted?

Reviewer 3 Report

It is an interesting manuscript describing the generative modeling of small molecules in a "Proteomic scale". The reviewer would say it is a joy reading this manuscript. The method development is solid and the result is well-presented and discussed. The following are some minor suggestions.

  1. For evaluating the generation outcome, Quantitative Estimation of Drug-likeness (QED) can be incorporated for assessment. https://www.ncbi.nlm.nih.gov/pmc/articles/PMC3524573/
  2. Besides the synthetic complexity, the assessment using the synthetic accessibility score (SAscore) can be one additional dimension to compare across groups. https://pubmed.ncbi.nlm.nih.gov/20298526/ 
  3. Benchmark models like GuacaMol, etc. could be compared for discussion. 
  4. Regarding the validation. The best way is adding wet-lab experiments. The reviewer is not expecting the synthesis and testing of dozens of derivatives for a comprehensive SAR, but at least several molecules for the proof-of-concept to see if generated molecules are really anti-aging or anti-cancer.

The reviewer is not an author to any paper mentioned above. Overall, the reviewer would suggest the minor revision.

Reviewer 4 Report

An interesting concept is put forward by the authors. I have few points to make. 

l. 205: What is meant by behavioral similarity? Is it toward specific target(s), or structural fingerprints, and/or physicochemical properties?

Synthetic accessibility based on molecular complexity and other reaction pathway features based on fragment sets are available in the literature. How does the SCScore method described here compares with the available methods? What is the advantage of it over the existing algorithms?

Being biased in predictive modeling is a very common feature, and authors are right to mention that. Can the predictive modeling be done heuristically or by implementing a dynamic library?
